# The Non-Affected Muscle Volume Compensates for the Partial Loss of Strength after Injection of Botulinum Toxin A

**DOI:** 10.3390/toxins15040267

**Published:** 2023-04-03

**Authors:** Reinald Brunner, Enrico De Pieri, Christian Wyss, Claudia Weidensteiner, Katrin Bracht-Schweizer, Jacqueline Romkes, Meritxell Garcia, Norine Ma, Erich Rutz

**Affiliations:** 1Department of Paediatric Orthopaedics, University Children’s Hospital Basel (UKBB), 4056 Basel, Switzerland; 2Laboratory of Movement Analysis, University Children’s Hospital Basel (UKBB), 4056 Basel, Switzerland; 3Department of Biomedical Engineering, University of Basel, 4123 Allschwil, Switzerland; 4Division of Radiological Physics, Department of Radiology, University Hospital Basel, 4031 Basel, Switzerland; 5Department of Neuroradiology, University Hospital Zürich, 8091 Zürich, Switzerland; 6Division of Neuroradiology, Clinic for Radiology & Nuclear Medicine, University Hospital Basel, 4031 Basel, Switzerland; 7Orthopaedic Department, The Royal Children’s Hospital, Melbourne 3052, Australia; 8Murdoch Children’s Research Institute—MCRI, Melbourne 3052, Australia; 9Department of Paediatrics, The University of Melbourne, Melbourne 3052, Australia; 10Medical Faculty, University of Basel, 4000 Basel, Switzerland

**Keywords:** cerebral palsy, botulinum toxin, muscle volume, efficacy, musculoskeletal modelling, MRI

## Abstract

Local botulinum toxin (BTX-A, Botox^®^) injection in overactive muscles is a standard treatment in patients with cerebral palsy. The effect is markedly reduced in children above the age of 6 to 7. One possible reason for this is the muscle volume affected by the drug. Nine patients (aged 11.5; 8.7–14.5 years) with cerebral palsy GMFCS I were treated with BTX-A for equinus gait at the gastrocnemii and soleus muscles. BTX-A was administered at one or two injection sites per muscle belly and with a maximum of 50 U per injection site. Physical examination, instrumented gait analysis, and musculoskeletal modelling were used to assess standard muscle parameters, kinematics, and kinetics during gait. Magnetic resonance imaging (MRI) was used to detect the affected muscle volume. All the measurements were carried out pre-, 6 weeks post-, and 12 weeks post-BTX-A. Between 9 and 15% of the muscle volume was affected by BTX-A. There was no effect on gait kinematics and kinetics after BTX-A injection, indicating that the overall kinetic demand placed on the plantar flexor muscles remained unchanged. BTX-A is an effective drug for inducing muscle weakness. However, in our patient cohort, the volume of the affected muscle section was limited, and the remaining non-affected parts were able to compensate for the weakened part of the muscle by taking over the kinetic demands associated with gait, thus not enabling a net functional effect in older children. We recommend distributing the drug over the whole muscle belly through multiple injection sites.

## 1. Introduction

Cerebral palsy (CP) is caused by a non-progressive lesion of the immature brain, occurring before, at, or shortly after birth. The resulting clinical picture is a motor disorder which impedes normal movement function and, therefore, affects participation in common life activities. The Gross Motor Function Classification System (GMFCS) can be used to clinically classify the patient-specific severity of functional impairment [1].

Motor symptoms commonly present themselves as a combination of both muscle weaknesses and increased or inadequate muscle activity. Inadequate muscle activity can impair the execution of common motor tasks, such as walking. Gait disorders are therefore a major reason for functional treatment. Spasticity is the most common type of inadequate muscle activity, in addition to dystonia and the less frequent ataxia. Spasticity is defined as an increased muscular resistance against a fast movement [2]. Two types of resistance of neural origin are distinguished: hyperreflexia and generally increased muscle tone [3]. The upper motor neuron syndrome, the underlying pathology in CP, is characterized by both spasticity with unmasked spinal reflexes and muscle weaknesses.

The occurrence of spastic or inadequate muscle activity over prolonged periods of time can lead to the development of contractures, structural muscle shortness, and eventually to fixed functional deformities, such as pes equinus, which further aggravate the functional severity of the motor disorder. When the overactivity of a single or a small number of muscles can be identified, weakening the responsible muscles is an attractive treatment for correcting the abnormal muscle activity, thus reducing the risk of developing fixed functional deformities. However, from a functional point of view, the spasticity-related stiffness of a muscle can functionally compensate for its weakness. During instrumented gait analysis, we observed that patients present less controlled and less timely uncoordinated muscle activity during walking. The presence of muscle shortness can lead to an overall stiffness of the muscle–tendon unit, which produces some resistance against gravity and the inertial forces acting on the body, thus passively supporting the lower limb during motion and preventing it from collapsing. Reducing muscle overactivity and spasticity could therefore unmask underlying muscle weaknesses; however, this runs the risk of further deteriorating overall motor function.

Structural muscle contractures can be surgically addressed, although surgery is usually avoided before the age of 6 or 7. The corrective shortening of the muscle–tendon complex is difficult and the results are poor. On the other hand, lengthening the muscle–tendon unit also bears the danger of further weakening the muscles, potentially leading to even poorer functional outcomes, such as crouch gait, for which treatment options are limited.

In contrast, the use of botulinum toxin A (BTX-A) enables the reduction of muscle overactivity in a temporary and reversible manner. The effect of BTX-A lasts long enough (longer than three months) to expose possible severe weaknesses which would further impair overall motor function. In this sense, BTX-A injections can be performed before surgery to rule out and avoid possible negative and detrimental effects of surgical lengthening [4]. The repetitive applications of BTX-A are reported to carry the risk of muscle damage [5,6] and hence do not represent an alternative to surgery. From a pharmacological point of view, BTX-A is a potent and reliable drug. Thus, the application of BTX-A has become a standard treatment to reduce local muscle hyperactivity and spasticity in the global regimen of functional treatments. The positive effects of BTX-A have been widely documented and consensus for the application of the drug, especially for the treatment of equinus deformities in younger children, has been published [7,8,9]. A maximal dose of 50 U per injection site for BTX-A was proposed [10]. However, during our routine clinical practice, we noticed a limited efficacy of BTX-A in the treatment of motor impairments and in restoring a normal gait function in patients. Our impression fits with more recent publications which throw some doubt on the reliability and efficacy of BTX-A [11,12,13,14], especially in children above the age of 6–7 [5]. The reason for this, however, is unclear.

While the mechanism of action of BTX-A has been scientifically investigated in detail [15,16,17,18], the distribution of the drug within the muscle tissue following injection and the behavior of the non-affected muscle tissue is less understood. In a first preliminary study, we investigated T2-weighted MRI in order to assess the muscle volume affected by the drug after injection [19]. We observed that a single BTX-A injection in the muscle bellies of gastrocnemii and soleus muscles affected 9–15% of the total muscle volume in a cohort of toe-walking children with CP who received BTX-A for pre-operative testing. This spatially limited hyperintensity on the T2-weighted images (detected up to 12 weeks post treatment) was presumably caused by an increase in the extracellular space around the atrophic muscle fibres. Most of the muscle volume remains therefore substantially unaffected. Hypothetically, this unaffected part of the muscle behaves in three different ways: it can become protracted and inefficient; it can contract only the same amount as before the injection, providing the same partial muscle force (while the total force would still be reduced due to the lack of contribution from the affected part); or it can generate a larger contractile force and therefore compensate for the effect of BTX-A on the affected part. The last option would not allow the unveiling and testing of the effect of a specific muscle weakness on the functionality of the patient.

The current study aims to characterize the biomechanics of walking and the motor function of a cohort of toe-walking children with CP before and after BTX-A injection where the affected muscle volume is known, by means of instrumented gait analysis and musculoskeletal modelling. In particular, the aim is to evaluate the kinetic requirements placed on the injected muscles during gait, to understand whether the partial weakening caused by BTX-A is sufficient to impair the functionality of these muscles, or if they still have a sufficient force-generating capacity. It is hypothesized that if only a small part of the muscle volume is affected by the injection, the patients would still have a sufficient residual muscle strength to meet the kinetic requirements associated with their usual pre-operative gait pattern and would therefore not change their walking pattern. The capacity of these muscles to produce a similar kinetic effect after injection would suggest that the non-affected muscle volume is able to functionally compensate for the local muscle weakness.

## 2. Results

### 2.1. BTX-A-Injection

Four patients had one site injected with 50 U BTX-A at each muscle belly of the triceps surae (2.8–4.6 U/Kg BW). In two patients, only the gastrocnemii were injected (two sites, 50 U/site; 5.6 and 8.8 U/Kg BW), and the others received injections at one or two sites for all muscle bellies (35–50 U/site; 5.4–8.5 U/Kg BW) (Table 1).

### 2.2. Clinical Examination

Only the reduction in spasticity 12 weeks after the BTX-A injection reached statistical significance (*p* = 0.034). All other clinical parameters at the different time points remained unchanged (Table 2 and Table 3). No statistically significant correlations between demographic factors and interventions regarding outcome measures were found. The linear regression analysis did not reveal any significant effect of the administered dose of BTX-A in the medial gastrocnemius on the range of ankle dorsiflexion with an extended knee and fixed subtalar joint.

### 2.3. MRI

Muscle volumes and the volume of the hyperintense regions of interests (ROIs), i.e., muscle volume affected by BTX-A, at a time point 6 weeks post-BTX-A injection are summarized in Table 4. As already discussed in [19], only parts of the treated muscles showed an MRI-visible effect after the BTX-A injection (Table 4).

### 2.4. Gait and Musculoskeletal Modelling Analysis

The patients presented altered pre-operative lower-limb kinematics compared to the control group (Figure 1).

Specifically, the gait of the patient was characterized by a significantly larger anterior pelvic tilt from mid-stance to initial swing (15–71% GC); a more flexed hip during terminal stance and mid-swing (50–55% and 82–92% GC); a more flexed knee during the swing phase (82–100% GC); a more plantarflexed ankle during initial contact, mid-to-terminal stance, and the swing phase (0–8%, 18–48%, and 67–100% GC); and a more flexed position of the foot compared to the floor during initial contact, mid-stance, and the swing phase (0–12%, 16–41%, and 70–100% GC). The SPM repeated-measures ANOVA analysis did not reveal any significant effect of BTX-A between pre-, 6 weeks post-, and 12 weeks post-operative lower limb kinematics.

Compared to controls, the patients presented significantly shorter pre-operative dynamic lengths of the gastrocnemius and soleus muscle–tendon units from mid-swing to initial contact (70–3% and 66–5% GC, respectively) (Figure 2).

At the same time, the contraction dynamics of the gastrocnemius were characterized by an elongation phase from terminal swing to initial foot contact (instead of shortening, 95–10% GC), an almost isometric phase during loading response (instead of elongation, 13–23% GC), by a reduced elongation during midswing (68–84% GC). The contraction dynamics of the soleus in the patients presented similar differences: 96–9% GC and 12–19% GC). No significant effect of BTX-A on the length and contraction velocity of the triceps surae muscle tendon units was found.

In terms of joint kinetics, the patients presented a significantly reduced internal knee extension moment during loading response and midswing (3–23% GC and 68–77% GC), as well as an increased plantarflexion moment during initial contact and loading response (0–19% GC) (Figure 3).

The knee sagittal power only presented limited differences in the transition from swing to stance, while the ankle sagittal power was characterized by an absorption phase during initial contact (0–11% GC) and by a decreased peak generation phase during push off (50–55% GC). Finally, the gait pattern of the patients translated into significantly different required muscle forces by the triceps surae compared to controls with higher forces during early stance (0–13% GC) and a reduced peak force during push-off (37–51% GC). No significant effect of BTX-A on sagittal joint moments, powers, or muscle forces was found.

## 3. Discussion

BTX-A has become a standard treatment tool for gait disorders in patients with CP. The mechanism of action of BTX-A has previously been studied and described in detail. By blocking neurotransmission at the motor end plate [15,16,17,18], BTX-A can be used to control inadequate and spastic muscle activity. Local muscle overactivity is reduced by injecting BTX-A into target muscles that impair the overall motor function of the patient. This treatment is often combined with corrective casts and/or intensive physiotherapy [20,21,22]. The clinical benefits have been documented in several papers [5,7,8,9,23]. Negative and dangerous side effects led to the current dose recommendations for the application of BTX-A [7,8]. However, BTX-A is not very effective for treating existing pes equinus deformities, especially in children over the age of 6. After this age, the pes equinus has often developed into a fixed deformity, leaving the surgical lengthening of the calf muscle–tendon unit as the most suitable treatment alternative [5]. As excessive muscle weakening is a dreaded complication of surgically lengthening a muscle–tendon unit (with poor options for subsequent correction), the application of BTX-A was introduced to temporarily reproduce weaknesses of specific muscles, thus revealing possible adverse consequences of these weaknesses on gait [4]. In this case, only the pure effect of BTX-A is desired and no other treatment is added. The nine patients involved in this study were 11.5 years old on average (range: 8.7–14.5), with six of them presenting a fixed equinus deformity. In the presence of a fixed deformity, BTX-A can only have an effect on the non-contracted part of the muscle, thus leading to a limited functional effect.

The present study did not focus on the effect of BTX-A per se but on the muscle volume affected by the drug within the muscle, as well as on the functional consequences that limited local injections induce on the remaining unaffected parts of the muscles. In a previous preliminary study investigating the same cohort of patients, we observed that only 9 to 15% of the muscle was affected with the present BTX-A application method [19]. These findings were confirmed in the additional four patients included in this study, and an overview of the results in all nine patients is presented here (Table 4). Hyperintensity on T2-weighted MRI scans were observed in all patients at the injection sites. The hyperintense area was spatially limited to parts of the injected muscles.

The current study also investigated the functional effect of the injection during gait, assessed by physical examination and instrumented gait analysis. The parameters of the physical examination (passive range of motion, MMST, modified Ashworth scale) remained mostly unchanged. A reduction in the manually assessed spasticity was the only indication of a minor positive effect of BTX-A.

Kinematic and net joint kinetic gait data did not reveal any effect of BTX-A on gait function. Similarly, musculoskeletal modelling indicated no functionally relevant effect of BTX-A during gait. Joint kinetics during gait did not show any significant difference after BTX-A injection, meaning that the overall kinetic demand placed on the plantar flexor muscles also remained unchanged (Figure 3). The triceps surae was still required to generate the same total force to meet the kinetic demands of the given kinematic pattern, which remained unchanged after the injection (Figure 1). This is in agreement with the findings of Wesseling et al., who described only a limited effect of BTX-A on muscle forces computed through musculoskeletal modelling in 14 children with spastic diplegic CP [24]. The loads that the triceps surae of the patients must sustain were significantly different from those of TD children. The toe walking gait pattern of the patients leads to an earlier and prolonged activation of these muscles, with higher forces during early stance and a reduced peak force (as well as a reduced peak ankle power) during push off [25].

As joint kinematics remained unchanged after injection, so did the dynamic length of gastrocnemius and soleus muscle–tendon units during gait (Figure 2). Compared to controls, the muscle–tendon units of the patients pre-BTX-A were shorter from mid-swing to heel strike, while no significant difference was found during loaded stance phase. This might indicate that any fixed contracture that the patients presented in the triceps surae was not sufficient for preventing a normal operating range of these muscles during gait. Furthermore, both muscle–tendon units were elongated during heel strike (from terminal stance to load acceptance), significantly differing from TD children, who presented either a shortening or a stationary length of these muscles in this phase of the gait cycle. This coincides with a phase of negative power and energy absorption at the ankle joint in patients when the muscles generated an eccentric force while elongating. A concomitant force production in an elongation phase might be linked to the activations of spastic reflexes; however, this would need to be investigated through specific targeted studies.

Overall, we observed a spatially limited effect of BTX-A on muscle tissue from the MRI analysis, while the functional demands placed on the triceps surae during gait did not change. Hence, since the muscles are generating the same total net force after injection, the non-affected part of the muscle must compensate for the partial and localized loss of muscle activity induced by BTX-A.

Reports of poor functional effects of BTX-A are rare [14,24,26]. We found a muscle volume effected by BTX-A in MRI without a functional change. Our findings fit with the description of Multani et al. who found BTX-A to be less effective in older children [5]. The mean age of the patient cohort of our study was 11.5 years, where the effect of BTX-A is reduced [5]. Our data suggest the muscle volume affected by a single site injection as a possible reason. Gough and Shortland also suggested increased muscle fibrosis as a possible factor for the loss of efficacy of BTX-A in older children [27,28,29,30,31]. Immunoresistance can be excluded as a reason for the lack of efficacy in our study, as an effected muscle volume was detected in MRI, although only in a small part of the treated muscle belly. A local immunoresistance, however, cannot be excluded. As we are extremely reluctant to the application of BTX-A, immunoresistance in all patients of this cohort is not probable. We applied BTX-A to one or two injection sites per muscle belly. This way of application did not produce a functionally relevant effect of BTX-A in our patients older than 8.5 years of age. This can be seen as a consequence of the low dose, the low number of injection sites, or of both. A possible solution may be to split up the dose into a larger number of injection sites to increase the affected muscle volume. Nevertheless, some doubts regarding the total injected volume persist. It is possible that, due to existing concerns for major adverse complications, the current recommended dose is too conservative and not sufficient to achieve a functional effect. BTX-A is more efficient in younger children [5]. This could be due to the fact that the total muscle volume is smaller in younger patients. A single injection site can therefore be sufficient to affect a relative larger volume of the muscle. Unfortunately, there is no evidence to support this hypothesis. Another possibility is fibrosis which develops over time in spastic muscles. Fibrosis could also hinder the diffusion of BTX-A but evidence is lacking. The need for anaesthesia during MRI investigations in younger children would also represent a further ethical issue for future investigations.

In addition, more questions arise: What would happen in multi-level applications, where the total drug volume has to be distributed in a larger number of muscles? How should more elongated muscles, such as hamstrings and psoas, be injected? Ideally, the whole muscle belly should be reached by the BTX-A treatment. In this sense, the number of injection sites seems to be more important than a locally applied dose, especially if the treatment is aimed at achieving only partial muscle weakening.

The patients included in this study were mostly male (seven male vs. two female). Lundkvist Josen by et al. found in a review paper that BTX-A was significantly more often applied in male patients, which confirms our distribution [32]. However, there are no clear reports on the effect of sex on the efficacy of BTX-A.

This study has some limitations, for instance, the fact that the number of patients is relatively low but all patients showed the same reaction. Additionally, the dose was low and the injection sites were few, in terms of the muscle size and according to current practice. Despite the different application modalities, there was no evident effect on the size of the affected muscle volume in our data. The small number of patients did not allow for the calculation of gender differences regarding muscle volume. Nevertheless, the current study did not focus on the overall effect of BTX-A but rather on the muscle volume affected by the drug within the muscle and the functional consequence of this partial affection of the muscle. The analysis of patients with CP through musculoskeletal modelling is also characterized by a series of limitations and assumptions. Conventional musculoskeletal modelling workflows rely on identifying an optimal solution for the muscle recruitment problem. However, the optimality of motor control in CP patients cannot be assumed. For this reason, our study focused on the geometrical outcomes of the musculoskeletal models, such as muscle–tendon unit lengths and velocities, whose values are dependent solely on musculoskeletal anatomy and joint kinematics, rather than on muscle recruitment solutions. In terms of kinetics, we analyzed the predicted muscle forces for the whole triceps surae group, rather than differentiating between gastrocnemius and soleus forces. In order to maintain a kinetic equilibrium of forces and moments around the joint ankle at every timepoint during the stance phase, a net ankle plantarflexing moment (and force) must be generated by the distal muscles. As the gastrocnemius and the soleus are the predominant plantar flexing muscles, the force they jointly exert is the major driver of ankle plantar flexion. While it is not possible to precisely identify how the load is shared across individual muscles in CP patients, the total force of the triceps surae group is a more reliable indicator of the kinetic demands placed on these muscles during gait. The maximum strength of these muscles was not personalized according to volumetric MRI data in our study but rather were scaled from the overall anthropometrics of each subject. Similarly, we did not model any weakening of the muscles after injection. While further personalization is always advisable, particularly for a better differentiation of gastrocnemius and soleus activity, the generic scaling of muscle strength we chose did not negatively impact the conclusions of this study. The fact that the patients were able to walk with similar joint kinematics and kinetics after injection confirms that they retained sufficient strength, similarly to the outcomes after modelling. Further studies should nevertheless investigate the effect of the more precise personalization of muscle volume, strength, and weakening to better understand the functional effect of BTX-A injections in individual muscles.

The local application of a moderate dose enabled us to answer the study question regarding the affected volume and overall response of the remaining part of the muscle. The low number of patients also represents a limitation for the statistical analysis of clinical parameters. The major reason for the low number of included patients was the existing restrictions due to COVID-19 regulations at the time of the study and the consequent dropouts.

In conclusion, the current recommendation of dosing BTX-A (Botox^®^) with one or two injection sites per muscle belly affects only a limited part of the muscle volume. The remaining unaffected part of the muscle volume is able to compensate for the weakened part of the muscle by taking over the kinetic demands associated with gait, thus not resulting in a net functional effect on gait in older children. Distributing the drug over the whole muscle belly by increasing the injection sites seems to be essential.

## 4. Methods

### 4.1. Ethical Considerations

Patients were recruited through a standard clinical procedure, and the effect of a possible muscle weakening after soft tissue surgery was simulated by the application of BTX-A in the respective muscles. Apart from MRI and a specific timepoint at 12 weeks, there was no additional load for the patients. Informed written consent was obtained from the participants’ parents and additionally from those participants aged 12 years or older. The study was approved by the local ethical committee (Ethikkommission Nordwest- und Zentralschweiz EKNZ 2016-01408, 15 October 2017). It is not possible to identify the individual patients using the data presented here.

### 4.2. Participants

Nine ambulatory patients with spastic CP (seven boys, two girls) were consecutively recruited from the preoperative BTX-A test routine before the surgical lengthening of the triceps surae-Achilles tendon complex for toe walking [4] and prospectively monitored between 2018 and 2021. Exclusion criteria were previous surgeries on the affected limb(s), claustrophobia, and difficulties in following instructions during the MRI investigation. The patients underwent an instrumented gait analysis and MRI prior to, 6 weeks after, and 12 weeks after BTX-A injection. One missing session was another exclusion criterion. Average age was 11.5 (range 8.7–14.5) years. Two patients had bilateral CP and seven had unilateral CP. All patients were classified as GMFCS 1. The patients with bilateral CP were asymmetrically affected and only one side required correction. Table 1 shows the detailed information on the patients. An age-matched cohort of 12 typically developing (TD) children (six boys and six girls, age: 11.5 [range 8.4–16.3] years) was included as a reference for comparison.

### 4.3. Treatment

We used BTX-A (Botox^®^, Allergan plc, Dublin, Ireland) and followed the manufacturer’s guidelines for the dose in lower limb application in children (https://media.allergan.com/actavis/actavis/media/allergan-pdf-documents/product-prescribing/20190620-BOTOX-100-and-200-Units-v3-0USPI1145-v2-0MG1145.pdf (accessed on 15 October 2017)): The total dose was limited to 300 U. We applied a maximum of 50 U in 1 mL NaCl per injection site, at a maximum of two sites per muscle belly. In order to avoid long travelling distances, BTX-A injections were administered at three different centres. BTX-A was administered according to the local slightly different standards but was always documented by ultrasound control (Table 1). As the effect on the different parts of the muscle belly and the possible compensation of the unaffected parts was the topic of research and not the efficacy of BTX-A, these differences were not considered important. No additional treatments, such as casting, additional physiotherapy, or additional/modified orthotics, were applied in order to solely investigate the effect of BTX-A injections on the whole muscle.

### 4.4. MRI

The MRI methods and MRI results of six of the ten patients were already published [19]. Briefly, the T2-weighted MR imaging of the leg muscles was performed at 3T with an accelerated acquisition method combined with model-based reconstruction (GRAPPATINI) [33]. This fast multi-echo spin echo sequence was used to acquire 32 axial slices with a voxel size of 1.0 mm × 1.0 mm × 3.0 mm, a repetition time TR of 4.3 s, and echo times TE ranging from 11 ms to 115 ms in 2 min 6 s. The image analysis was performed on the T2-weighted images with TE = 34 ms. Hyperintense regions were visible 6- and 12-weeks post-BTX-A injection. The hyperintense region at a timepoint of 6 weeks was segmented and defined the muscle volume affected by the drug. Additionally, the gastrocnemius medialis and lateralis and soleus muscles were segmented in both legs and their volume was calculated.

### 4.5. Gait Analysis

The patients underwent kinematic (motion capture system: Vicon Motion Systems Ltd., Oxford, UK) and kinetic (force platforms: Kistler Group, Winterthur, CH) 3D gait analysis before BTX-A injection and 6 weeks (6.1 [range 5.4–6.9] weeks) and 12 weeks (12.3 [range 11.4–12.9] weeks) after BTX-A injection. TD children participated in a single measurement session. Gait data were measured barefoot at a self-selected speed using the Plug-in Gait lower-body marker-set [34] until six valid kinetic datasets for each leg were acquired. During each clinical gait analysis, a physical examination was performed, including a test of the passive range of motion, a manual muscle strength test (MMST), and an assessment of spasticity (modified Ashworth scale).

### 4.6. Musculoskeletal Modelling

Marker trajectories and ground reaction force (GRFs) data were used as input for an inverse dynamics analysis in the AnyBody Modeling System (AnyBody Technology, Denmark) [35]. Individual models for each subject were created from a detailed musculoskeletal model of the lower limb [36,37], which was scaled to match the overall anthropometrics and the marker data collected during a standing reference trial [38]. The hip joints were modelled as three-degrees of freedom of (DoF) ball-and-socket joints, while knee, talocrural, and subtalar joints were modelled as one-DoF hinges. The muscle elements were modelled with a simple muscle model represented by constant strength actuators.

Sagittal plane joint kinematics were computed from the measured marker trajectories and reported according to the International Society for Biomechanics’ (ISB) recommendations [39]. Pelvic tilt and foot–floor inclination angle were computed with respect to the laboratory reference frame. An inverse dynamics analysis based on a third-order-polynomial muscle recruitment criterion was then performed to calculate required muscle activations and forces, as well as the resulting internal net sagittal moments at the knee and ankle joints [40]. Joint power in the sagittal plane was calculated as a product of the net sagittal moment times the rotational velocity of the joint. The total force required by the triceps surae to reproduce the given motion was computed [25]. Finally, the instantaneous dynamic length of the muscle–tendon units during gait, as well as their shortening or elongation velocity, were analyzed [41,42,43,44]. Muscle lengths were normalized as a percentage of the length of the tibia for each patient.

All gait trials were processed and analyzed through the toolkit AnyPyTools [45] in the Python programming language (Python Software Foundation, Wilmington, DE, USA). The analysis was limited to the affected and injected leg of the patients, while a randomly chosen leg was analyzed for the control group. Joint moments, powers, and muscle forces were normalized to body weight (BW), while muscle lengths were normalized to the length of the tibia, which was calculated during the above-mentioned scaling of the musculoskeletal models starting from markers’ data. Average trajectories per subject were then calculated based on the collected walking trials. All the data were time-normalized to the gait cycle (GC) from the foot-strike (0%) to foot-strike (100%) of the leg of interest.

### 4.7. Statistical Analysis

Outcome measures were compared between pre-intervention at two different time points (comparing pre-intervention to 6- and 12-weeks follow-up) using a paired *t*-test when data were normally distributed and a Wilcoxon’s signed ranked test on non-normal data distribution. Correlations between demographic factors and interventions to outcome measures were calculated using Pearson’s (normal distributions) or Kendall’s (non-normal) correlation. Linear regression was calculated between a range of the motion of dorsiflexion with extended knee and fixed subtalar joints and a dose of botulinum toxin administered to medial gastrocnemius.

Joint kinematics, moments, and powers, as well as muscle lengths, velocities, and forces, were analyzed using statistical parametric mapping (SPM; www.spm1D.org accessed on 17 March 2023, v0.43) [37]. Differences between controls and patients’ preoperative data were analyzed using nonparametric [38] two-tailed two samples *t*-tests. Nonparametric tests were chosen as conservative approach, considering the size of the cohort. Furthermore, Shapiro−Wilk tests confirmed that not all data were normally distributed. To assess the effect of BTX-A on the gait of patients, a repeated-measures ANOVA was carried out for each gait variable of interest, comparing pre-, 6 weeks post-, and 12 weeks post-operative data and allowing the differentiation of between-subject and within-subject effects, with the within-subject effect representing the BTX-A injection. The significance level was set a priori at α = 0.05 across all tests.

## Figures and Tables

**Figure 1 toxins-15-00267-f001:**
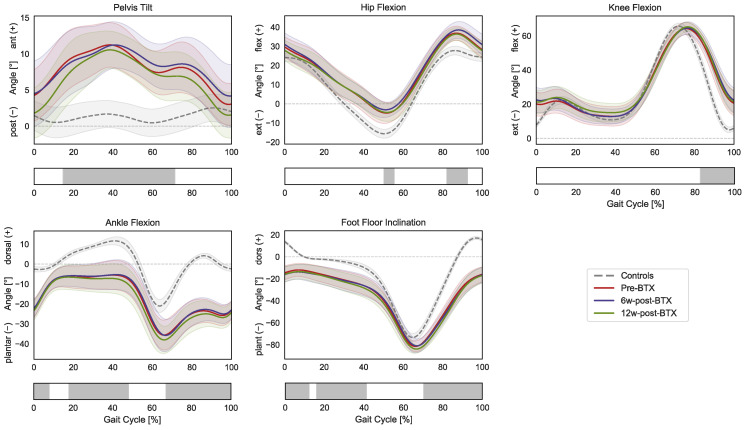
Lower-limb sagittal plane kinematics during the gait cycle (GC). Mean pelvis tilt (+ anterior, − posterior), hip flexion (+ flexion, − extension), knee flexion (+ flexion, − extension), ankle flexion (+ dorsiflexion, − plantarflexion), and foot–floor inclination angles (+ dorsiflexion, − plantarflexion), are reported in red for patients before injection (pre-BTX), in blue at 6 weeks after injection (6w-post-BTX), in green at 12 weeks after injection (12w-post-BTX). Reference data for the control subjects are reported as dashed grey lines. ±95% confidence intervals are indicated as shaded areas. Periods of GC with a statistically significant difference between controls and patients pre-BTX in the relevant SPM *t*-tests are indicated as grey bars below each subplot. No significant differences between pre-BTX, 6w-post-BTX, and 12w-post-BTX were found.

**Figure 2 toxins-15-00267-f002:**
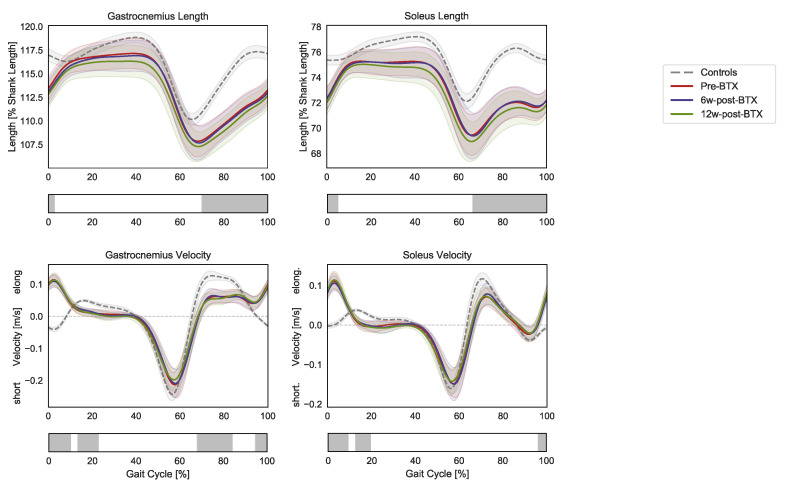
Lower-limb sagittal plane kinematics during the gait cycle (GC). Mean pelvis tilt (+ anterior, − posterior), hip flexion (+ flexion, − extension), knee flexion (+ flexion, − extension), ankle flexion (+ dorsiflexion, − plantarflexion), and foot–floor inclination angles (+ dorsiflexion, − plantarflexion) are reported in red for patients before injection (pre-BTX), in blue at 6 weeks after injection (6w-post-BTX), and in green at 12 weeks after injection (12w-post-BTX). Reference data for the control subjects are reported as dashed grey lines. ±95% confidence intervals are indicated as shaded areas. Periods of GC with a statistically significant difference between controls and patients pre-BTX in the relevant SPM *t*-tests are indicated as grey bars below each subplot. No significant differences between pre-BTX, 6w-post-BTX, and 12w-post-BTX were found.

**Figure 3 toxins-15-00267-f003:**
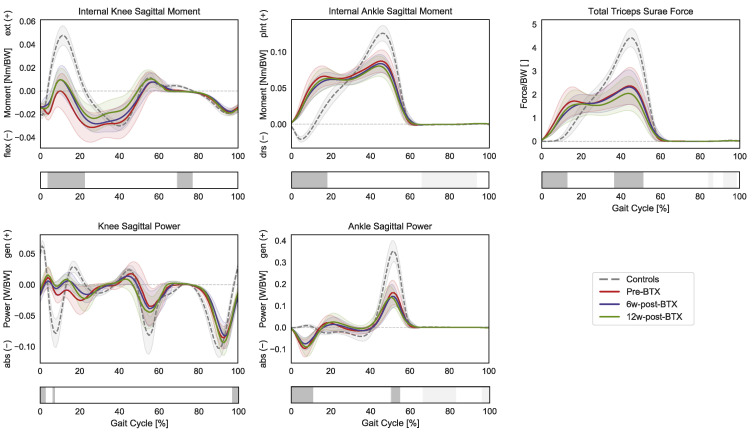
Distal lower-limb kinetics during the gait cycle (GC). Mean knee internal sagittal moment (+ muscle extension, − muscle flexion), ankle internal sagittal moment (+ muscle plantarflexion, − muscle dorsiflexion), and knee and angle sagittal powers (+ generation, − absorption), as well as the total force exerted by the triceps surae (gastrocnemius and soleus), are reported in red for patients before injection (pre-BTX), in blue at 6 weeks after injection (6w-post-BTX), and in green at 12 weeks after injection (12w-post-BTX). Reference data for the control subjects are reported as dashed grey lines. ±95% confidence intervals are indicated as shaded areas. Periods of GC with a statistically significant difference between controls and patients pre-BTX in the relevant SPM *t*-tests are indicated as grey bars below each subplot. No significant differences between pre-BTX, 6w-post-BTX, and 12w-post-BTX were found.

**Table 1 toxins-15-00267-t001:** Patient overview: sex, type of CP involvement, dosage, number of sites, and location of BTX-A injections.

			BTX-A Injection in Calf Muscles
Patient Number	Sex	Type of CP	Units/Site and Muscles Injected
1	f	unilateral	50 U, 1 site/muscle belly, gastroc. med. and lat. and soleus right
3	m	unilateral	50 U, 1 site/muscle belly, gastroc. med. and lat. and soleus right
4	m	unilateral	50 U, 1 site/muscle belly, gastroc. med. and lat. and soleus right
5	m	bilateral	75 U, 2 sites/muscle belly, gastroc. med. and lat. and soleus left
6	m	bilateral	100 U, 2 sites/muscle belly, gastroc. med. and lat. left
7	m	unilateral	100 U, 2 sites/muscle belly, gastroc. med. and lat.; 50 U, 1 site, soleus left
10	f	unilateral	100 U, 2 sites/muscle belly, gastroc. med. and lat. and soleus left
11	m	unilateral	75 U, 2 sites/muscle belly, gastroc. med. and lat. right
12	m	unilateral	50 U, 1 site/muscle belly, gastroc. med. and lat. right

**Table 2 toxins-15-00267-t002:** Patient overview. Assessments from clinical examination.

**Pre-Btx**						
		**Spasticity** **(Modif. Ashworth)** **(Score)**		**ROM Dorsiflexion, with Extended Knee (Limiting Muscle)** **(Degrees)**		**Manual Muscle Strength Test** **(MMST)** **(Score)**
**Patient Number**		**Plantar Flexors, Knee at 90°**	**Plantar Flexors, Knee at 0°**	**Gastroc.**	**Soleus**	**Plantar Flexors**
1		1	1.5	−10	−15	2.5
3		1.5	1.5	−15	−30	2.5
4		0	1	−25	−25	2.5
5		2	3	−25	−30	3
6		1	1	−30	−20	3.5
7		1	1.5	0	0	2
10		0	0	−10	−15	1.5
11		1.5	1.5	5	5	3
12		0	0	5	5	2
Mean		0.9	1.2	−12.0	−14.0	2.5
Stand. Dev.		0.7	0.8	11.9	12.6	0.5
**6 Weeks after Btx**						
		**Spasticity** **(Modif. Ashworth)** **(Score)**		**ROM Dorsiflexion, with Extended Knee (Limiting Muscle)** **(Degrees)**		**Manual Muscle Strength Test** **(MMST)** **(Score)**
**Patient Number**	**Weeks after Btx**	**Plantar Flexors, Knee at 90°**	**Plantar Flexors, Knee at 0°**	**Gastroc.**	**Soleus**	**Plantar Flexors**
1	6.9	1	1	−10	−15	2.5
3	6.4	1.5	1	−15	−20	2.5
4	6.3	1	1	−20	−20	2.5
5	5.4	2	1	−20	−25	2
6	5.4	1.5	1.5	−20	−30	2.5
7	5.4	0	1	−10	−15	2
10	6.3	0	0	−25	−25	2
11	5.3	1	1	−5	−5	4
12	6.0	1	0	5	0	2
Mean	6.0	0.9	0.8	−13.0	−17.0	2.5
Stand. Dev.	0.6	0.7	0.5	8.4	8.7	0.6
**12 Weeks after Btx**						
		**Spasticity** **(Modif. Ashworth)** **(Score)**		**ROM Dorsiflexion, with Extended Knee (Limiting Muscle)** **(Degrees)**		**Manual Muscle Strength Test** **(MMST)** **(Score)**
**Patient Number**	**Weeks after Btx**	**Plantar Flexors, Knee at 90°**	**Plantar Flexors, Knee at 0°**	**Gastroc.**	**Soleus**	**Plantar Flexors**
1	12.9	1	1	−5	−5	3
3	12.4	1	1.5	−20	−25	2.5
4	12.3	1	0	−20	−25	2.5
5	12.3	1	1.5	−20	−25	2.5
6	11.4	1.5	0	−25	−30	2.5
7	12.0	0	0	−10	−15	2
10	12.4	0	0	−20	−30	2
11	12.3	1.5	1.5	5	0	4
12	11.9	0	0	−5	5	2.5
Mean	12.3	0.8	0.6	−13.5	−17.0	2.6
Stand. Dev.	0.4	0.6	0.7	9.0	12.1	0.5

**Table 3 toxins-15-00267-t003:** Statistical comparison between pre-, 6 weeks post-, and 12 weeks post-BTX-A outcome measures. *p*-values were computed using paired *t*-tests (normally distributed data) or Wilcoxon’s signed rank test (non-normally distributed data). Statistically significant differences (α < 0.05) are reported in bold. The values of ROM dorsiflexion (DF) with the knee extended are presented in Table 2.

	Pre-Botox vs. 6 Week F/U	6 Week F/U vs. 12 Week F/U	Pre-Botox vs. 12 Week F/U
Spasticity—PF with 90° kneeflexion	1	0.6193	0.6193
Spasticity—PF with extended knee	0.0676	0.3951	0.03351
ROM—DF with 90° kneeflexion and fixed subtalar joint	0.6681	0.8605	0.4273
ROM—DF with extended knee and fixed subtalar joint	0.2543	1	0.3198
ROM—DF with extended knee and free subtalar joint	0.7959	0.2795	0.4627
ROM—KE with extended hip	0.5716	0.7498	0.7728
Force MMST—PF	0.8501	0.08113	0.5911
Force MMST—KF	0.4679	0.7103	0.1382

**Table 4 toxins-15-00267-t004:** Muscle parameters from MRI. Muscle volumes of triceps surae of both legs as assessed by MRI, the volume of treated muscles, the volume affected by BTX-A (hyperintense ROI on T2-weighted MRI images), and relative ROI size (hyperintense ROI divided by the volume of treated muscles). All results were assessed on MRI data at timepoint 6 weeks post-BTX-A injection.

Patient Number	Treated Leg	Treated Muscles	Total Volume of Treated Muscles [cm³]	Volume of Hyperintense ROI [cm³]	Percentage of Hyperintense Tissue = Relative ROI Size	Total Volume of Triceps Surae [cm³]	Total Triceps Surae Volume of Contralateral Leg [cm³]
1	r	Gl+Gm+S	231	32.7	14%	231	393
3	r	Gl+Gm+S	225	19.6	9%	225	361
4	r	Gl+Gm+S	223	32.2	14%	223	400
5	l	Gl+Gm+S	203	21.3	10%	203	206
6	l	Gl+Gm	58	25.2	43%	124	196
7	l	Gl+Gm+S	249	34.8	14%	249	354
10	l	Gl+Gm+S	289	31.1	11%	289	478
11	r	Gl+Gm	116	11.1	10%	286	341
12	r	Gl+Gm	146	22.0	15%	335	495

## Data Availability

There is no dataset available for privacy reasons.

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
