# Peer review of "The Non-Affected Muscle Volume Compensates for the Partial Loss of Strength after Injection of Botulinum Toxin A"

_toxins, 2023, doi:10.3390/toxins15040267_

Round 1
Reviewer 1 Report
This paper investigates BTX-A area of effect after injections using MRI, and has implications for treatment of cerebral palsy.
Gait analysis had shown little to no benefit from BTX-A, leading the authors to study the amount of muscle effected by the drug, with the hypothesis that using the drug at multiple injection sites would be beneficial.
The overall conclusion that more injection sites could be used is reasonable and supported by the data. Proper statistical analysis was performed throughout, and I see no reason why the study should not be published.
Author Response
Thank you for your kind comments.
Reviewer 2 Report
Recommendations:
Line 22 – explain the data included in the reference as a part of another study, pointing out at the end of the paragraph the reference
Add ethical consideration section
Add limitations of the study and discuss the number of the participants and the heterogeneity of the group
Edit table 1 and make it readable
Author Response
Line 22 – explain the data included in the reference as a part of another study, pointing out at the end of the paragraph the reference.
Line 22 in our manuscript is the second line of the introduction. There is no data and no reference. The introduction part is re-written. We hope, this unclear issue is corrected this way.
Add ethical consideration section.
We have added the necessary information concerning the ethical issue. The paragraph reads now:
The patients were recruited from a standard clinical procedure where the effect of a possible muscle weakening after soft tissue surgery was simulated by the application of BTX-A in the respective muscles. Apart from MRI and a timepoint at 12 weeks, there was no additional load for the patients. Informed written consent was obtained from the participants’ parents and additionally from those participants aged 12 years or older. The study was approved by the local ethical committee (Ethikkommission Nordwest- und Zentralschweiz EKNZ 2016-01408, 15.10.2017). By the data currently presented it is not possible to identify the individual patient
Add limitations of the study and discuss the number of the participants and the heterogeneity of the group.
We added a discussion part on gender and on the effect on smaller children. We hope, this fulfils the request of reviewer 2. As the discussion part is re-written, we cannot point out the new sentences as usual if the manuscript is mostly unchanged. We added a comment on gender distribution as requested by reviewer 2 and 3. It may be altered due to re-writing and language editing.
The patients included in this study were mostly males (7 males vs 2 females). Lundkvist Josen by et al. found in a review paper that BTX-A was significantly more often applied in males which confirms our distribution [45]. However, there are no clear reports on the effect of sex on the efficacy of BTX-A.
Edit table 1 and make it readable.
We have modified Table 1 as suggested, and separated it into Table 1 and Table 2. We think, it is much clearer now.
Reviewer 3 Report
The manuscript described a small clinical study using botulinum toxin A for toe-walking children with cerebral palsy (age range 8.7 to 14.5 with an average of 11.5). The study is focused on the functional consequences of the muscle not affected by BTX-A. The distribution of BTX-A within the muscle (the muscle volume affected by BTX-A) was monitored using MRI. Overall, the muscle volume affected by BTX-A is small, and no effects in kinematic and kinetic conventional gait data, as well as in the musculoskeletal modeling parameters. Those results are in line with other studies that BTX-A is less effective in older children. The less effectiveness may be due to that the volume of affected muscle in this study is small and the remaining non-affected parts of the muscle compensate functionally and suggested using a higher dose in older children. It has been reported the less effectiveness of BTX-A in older children similar to the age group in this study, therefore, authors should make some discussions on the effectiveness on younger children. Comparing the volume of muscle affected by BTX-A in younger children may provide additional support for their study. Further, it seems to have a sex imbalance in their study with only 2 females. Their reference group with typically developing children is well balanced with 6 males and 6 females. They should discuss if this imbalance in sex will generate bias in their results. Some English expressions in the manuscript are also difficult to understand. It would be better to proofread the manuscript carefully.
Author Response
It has been reported the less effectiveness of BTX-A in older children similar to the age group in this study, therefore, authors should make some discussions on the effectiveness on younger children. Comparing the volume of muscle affected by BTX-A in younger children may provide additional support for their study.
We added a discussion part on the effect in younger children.
BTX-A is more efficient in younger children [12]. This could be due to the fact that the total muscle volume is smaller in younger patients. A single injection site can therefore be sufficient to distribute the drug in a relative larger volume of the muscle. Unfortu-nately, there is no evidence to support this hypothesis. The need for anaesthesia during MRI investigations in younger children would also represent a further ethical issue for future investigations.
Further, it seems to have a sex imbalance in their study with only 2 females. Their reference group with typically developing children is well balanced with 6 males and 6 females. They should discuss if this imbalance in sex will generate bias in their results.
We added a discussion part on the sex imbalance as described above (Reviewer 2)
Some English expressions in the manuscript are also difficult to understand. It would be better to proofread the manuscript carefully.
We re-submitted our paper and asked for a language check by MDPI. Hence, wording may be different.
Round 2
Reviewer 3 Report
The authors used the terminology “distribution” of BoNT-A in the muscle throughout the manuscript. However, the distribution described here is based on the muscle volume detected by MRI, not the conventional distribution in pharmacology, since no BoNT-A is directly detected in the muscle. To avoid confusion, recommend the authors change the terminology to affected muscle volume by BoNT-A, rather than distribution. Other revisions are fine.
Author Response
We thank you for your comment on the distribution of BTX-A. We agree that the term “affected muscle volume” is more appropriate and we changed the manuscript accordingly. All changes in the manuscript are marked in red.